# A Variant of Anderson Mixing
# with Minimal Memory Size

**Fuchao Wei[1], Chenglong Bao[3,4][*], Yang Liu[1,2], Guangwen Yang[1]**
[1]Department of Computer Science and Technology, Tsinghua University, China
[2]Institute for AI Industry Research (AIR), Tsinghua University, China
[3]Yau Mathematical Sciences Center, Tsinghua University, China
[4]Yanqi Lake Beijing Institute of Mathematical Sciences and Applications

## Abstract

Anderson mixing (AM) is a useful method that can accelerate fixed-point iterations by exploring the information from historical iterations. Despite its numerical success in various applications, the memory requirement in AM remains a bottleneck when solving large-scale optimization problems in a resource-limited machine. To address this problem, we propose a novel variant of AM method, called Min-AM, by storing only one vector pair, that is the minimal memory size requirement in AM. Our method forms a symmetric approximation to the inverse Hessian matrix and is proved to be equivalent to the full-memory Type-I AM for solving strongly convex quadratic optimization. Moreover, for general nonlinear optimization problems, we establish the convergence properties of Min-AM under reasonable assumptions and show that the mixing parameters can be adaptively chosen by estimating the eigenvalues of the Hessian. Finally, we extend Min-AM to solve stochastic programming problems. Experimental results on logistic regression and network training problems validate the effectiveness of the proposed Min-AM.

## 1 Introduction

Anderson mixing (AM) [2] is a powerful method for accelerating fixed-point iterations, and has been widely used in scientific computing [1, 5, 47] and machine learning [31, 58, 29]. In each iteration, AM extrapolates a new iterate that satisfies certain optimality property by using the historical iterations, and exhibits empirical acceleration over classical fixed-point iterations [60]. Thus, AM can reduce the computation cost when evaluating the fixed-point map is a time-consuming process. Also, the inspiring connections of AM with GMRES [60, 48] and multisecant quasi-Newton methods [22] provide some explanations for the acceleration benefit from AM. In recent years, there have been many works applying AM to solve various optimization problems [56, 62, 41]. Their promising results in different tasks suggest that these AM-based methods are very competitive while being more economical than Newton's method. However, the existing convergence theory of AM is still unsatisfactory [3] and AM can diverge in some case [41]. Consequently, the applications of AM and its variants for machine learning problems and the corresponding theories deserve deep research.

One major concern of AM is the heavier memory overhead compared with the fixed-point iteration. In AM($m$), it has to store vector pairs about historical information from the $m$ previous iterations, where $m$ is the memory size. Specifically, AM($m$) is called as the limited-memory AM if $m < \infty$, and full-memory AM if $m = \infty$. The AM(0) is the simple fixed-point iteration. In practice, choosing proper memory size $m$ is heuristic and it is observed in [47] that a small $m$ can deteriorate the efficacy. However, for practical applications, a large $m$ can lead to memory issue for solving large-scale and high-dimensional problems when the memory resource is limited.

---

[*]Corresponding author. clbao@mail.tsinghua.edu.cn.

36th Conference on Neural Information Processing Systems (NeurIPS 2022).

In this paper, we address the memory issue of AM for solving optimization problems by proposing Min-AM, a novel variant of AM(1) that has minimal memory size in AM. Min-AM is built upon the Type-I AM [22, 60], a variant of AM. Let AM-I($m$) denote the Type-I AM with memory size $m$. Using a projection-mixing framework of AM-I, the Min-AM retains a recursively modified vector pair and adds one extra projection step in each step of AM-I(1). Compared to AM(1), Min-AM forms symmetric approximations to the Hessian and is equivalent to AM-I($\infty$) when the objective function is strongly convex quadratic. By incorporating the restarting strategy, the restarted Min-AM has faster convergence rate than AM(1) for general nonlinear optimization and the spectrum of Hessian can be cheaply estimated that leads to adaptive choices of the mixing parameters. Min-AM is also extended to stochastic optimization. In summary, we highlight the main contributions of this work as follows.

- We propose Min-AM, a variant of AM(1) that achieves the minimal memory size in AM, for solving optimization problems. Compared to AM(1), Min-AM forms symmetric approximations to the Hessian and incorporates more historical information to obtain the update. Thus, Min-AM can significantly reduce the memory requirement and computational cost of AM while maintaining the fast convergence property.

- We show that the local convergence rate of the restarted Min-AM has optimal dependence on the condition number of the objective function. We provide adaptive choices of the mixing parameters by estimating the spectrum of the Hessian economically in each iteration. Also, we give the convergence and complexity analysis of a stochastic extension of Min-AM for solving nonconvex stochastic optimization.

- We verify the properties of Min-AM, and apply the restarted Min-AM to logistic regression and the stochastic Min-AM to train neural networks. The experimental results are consistent with our theoretical analysis and show that the Min-AM is competitive with the limited-memory quasi-Newton methods with large memory size in deterministic optimization and achieves promising results in training neural networks.

**Related work.** Fixed-point problems can be recast as solving systems of (non)linear equations. AM is a practical alternate for Newton's method when the Jacobian is unavailable or difficult to compute [35]. In the linear case, it is established in [60, 48] that the full-memory AM is equivalent to GMRES [51], a classical Krylov subspace method for linear systems. In the nonlinear case, AM is recognized as a multisecant quasi-Newton method and a variant, called Type-I AM, has been proposed in [22]. From the perspective of convergence analysis, it is proved that the local convergence rate of the limited-memory AM is no worse than that of fixed-point iteration [59, 16] and the potential improvement depends on the quality of extrapolation [21, 46]. However, a counter example [41] shows the divergence behaviour of AM(1) and the recent review [3] indicates the need of further research on the theoretical properties of AM.

The inspiring performance and the easy implementation of AM motivate many new variants of AM for various applications in machine learning. In [55], a regularized variant of AM is proposed to accelerate gradient descent for unconstrained optimization, which is adapted in [23] for nonsmooth convex optimization and in [58] for reinforcement learning. In [62], a stochastic version of AM is developed and has convergence guarantee for stochastic optimization. The Type-I AM is modified in [67] to solve nonsmooth fixed-point problems. A more recent work, the short-term recurrence AM (ST-AM) [63], stores two vector pairs to reduce the memory cost of AM. Our Min-AM method achieves the minimal memory size, i.e. $m = 1$, further reducing the memory size of ST-AM.

The memory overhead is a common issue of quasi-Newton methods. It is desirable to keep memory size as small as possible without sacrificing the efficacy of the algorithm. Some works [37, 9, 8] study the choice of memory size in L-BFGS [39], but the performance with minimal memory size is unsatisfactory. In stochastic programming, the sublinear convergence rate is optimal when only stochastic gradients can be accessed [43]. Thus the heavier memory and computational cost makes quasi-Newton methods [12, 27, 61, 10] less appealing than the more economical first-order methods [49, 36]. In this work, our proposed Min-AM method exhibits the fast convergence as quasi-Newton methods while the memory cost is close to first-order methods.

*Notations.* Let $\Delta$ denote the forward difference operator, e.g. $\Delta x_k = x_{k+1} - x_k$. range($X$) denotes the subspace spanned by the columns of a matrix $X$. For every matrix $A \in \mathbb{R}^{d \times d}$, the Krylov subspace $\mathcal{K}_m(A, v) \equiv \mathrm{span}\{v, Av, \ldots, A^{m-1}v\}$, and $\|x\|_A = (x^\mathrm{T} A x)^{1/2}$ denotes the $A$-norm when $A$ is symmetric positive definite (SPD). "†" is the Penrose-Moore inverse.

## 2 The Anderson mixing scheme

In this work, we consider the minimization problem

$$\min f(x), \ x \in \mathbb{R}^d, \tag{1}$$

where $f : \mathbb{R}^d \to \mathbb{R}$ is continously differentiable. The first-order optimality condition of (1) is equivalent to $x = g(x)$, where $g(x) = x - \nabla f(x)$. We describe the Anderson mixing scheme with the projection-mixing framework [62]. Define $r_k = g(x_k) - x_k = -\nabla f(x_k)$ to be the residual. AM($m$) maintains $m$ ($m \le \min\{d, k\}$) vector pairs stored in $X_k, R_k \in \mathbb{R}^{d \times m}$:

$$X_k = [\Delta x_{k-m}, \Delta x_{k-m+1}, \cdots, \Delta x_{k-1}], \ \ R_k = [\Delta r_{k-m}, \Delta r_{k-m+1}, \cdots, \Delta r_{k-1}]. \tag{2}$$

Each update can be decoupled into two steps, namely the *projection step* and the *mixing step*:

$$\bar{x}_k = x_k - X_k \Gamma_k, \quad \text{(Projection step)}, \ \ x_{k+1} = \bar{x}_k + \beta_k \bar{r}_k, \quad \text{(Mixing step)}, \tag{3}$$

where $\bar{r}_k := r_k - R_k \Gamma_k$, $\beta_k > 0$ is the mixing parameter, and $\Gamma_k = \arg\min_{\Gamma \in \mathbb{R}^m} \|r_k - R_k \Gamma\|_2$. It is easy to know that the Galerkin's projection condition $\bar{r}_k \perp \text{range}(R_k)$ holds. Assuming $R_k^{\mathrm{T}} R_k$ is nonsingular, the resulting one-step update of AM($m$) is $x_{k+1} = x_k + H_k r_k$ where

$$H_k = \beta_k I - (X_k + \beta_k R_k)(R_k^{\mathrm{T}} R_k)^{-1} R_k^{\mathrm{T}}. \tag{4}$$

The AM-I [22, 60] falls into the same projection-mixing framework, and the only difference lies in the choice of $\Gamma_k$. The $\Gamma_k$ of AM-I is determined by another Galerkin's condition:

$$\bar{r}_k^{\mathrm{T}} X_k = (r_k - R_k \Gamma_k)^{\mathrm{T}} X_k = 0. \tag{5}$$

Substituting the solution of (5) into (3), the approximated inverse Hessian matrix $H_k$ of AM-I is

$$H_k = \beta_k I - (X_k + \beta_k R_k)(X_k^{\mathrm{T}} R_k)^{-1} X_k^{\mathrm{T}}, \tag{6}$$

by assuming $X_k^{\mathrm{T}} R_k$ is nonsingular. AM-I has a close relation to Newton's method and an alternative interpretation of AM-I is given in Appendix A.1.

**Remark 1.** *It is established in [22] that the $H_k$ of AM solves $\min_{H_k} \|H_k - \beta_k I\|_F$ s.t. $H_k R_k = -X_k$ and the $H_k$ of AM-I satisfies $H_k = B_k^{-1}$, where $B_k$ solves $\min_{B_k} \|B_k - \beta_k^{-1} I\|_F$ s.t. $B_k X_k = -R_k$. Thus, both AM and AM-I are multisecant methods. Besides, from (4) and (6), the approximated inverse Hessian matrices $H_k$ are generally not symmetric.*

## 3 The Min-AM methods

In each iteration, AM($m$) has to store two matrices $X_k, R_k \in \mathbb{R}^{d \times m}$, which dramatically increases the memory burden in large-scale problems. To reduce the memory requirement, we consider the minimal memory case, i.e. $m = 1$. The proposed Min-AM is a variant of AM(1) and the convergence of Min-AM is established in this section.

### 3.1 The basic Min-AM

In this section, we consider the simple case of (1): the objective function is $f(x) = \frac{1}{2} x^{\mathrm{T}} A x - b^{\mathrm{T}} x$ where $A$ is SPD. Let $s_k := \Delta x_{k-1}, y_k := \Delta r_{k-1}$, and choose $m = 1$ in (4). The approximated inverse Hessian of AM(1) is $H_k = \beta_k I - (s_k + \beta_k y_k)(y_k^{\mathrm{T}} y_k)^{-1} y_k^{\mathrm{T}}$, which is non-symmetric. However, the Hessian matrix $\nabla^2 f$ is naturally symmetric. To achieve a better approximation of $\nabla^2 f$, we first recursively modify the vector pair $s_k, y_k$ to incorporate more information from the previous iterations. Let $p_1 = \Delta x_0, q_1 = \Delta r_0 \in \mathbb{R}^d$. For $k \ge 2$, we construct $p_k$ and $q_k$ by

$$p_k = \Delta x_{k-1} - p_{k-1} \zeta_k, \quad q_k = \Delta r_{k-1} - q_{k-1} \zeta_k, \tag{7}$$

where $\zeta_k = (p_{k-1}^{\mathrm{T}} q_{k-1})^{-1} p_{k-1}^{\mathrm{T}} \Delta r_{k-1}$, assuming $p_{k-1}^{\mathrm{T}} q_{k-1} \ne 0$. Then, inspired by the Two-Grid Cycle method [53, Algorithm 13.2] in the multigrid techniques [11], we add an extra projection step to (3), and the resulting scheme is

$$x_k^{(1)} = x_k - p_k \Gamma_k^{(1)}, \qquad \text{(Projection step)} \tag{8a}$$

$$x_k^{(2)} = x_k^{(1)} + \beta_k r_k^{(1)}, \qquad \text{(Mixing step)} \tag{8b}$$

$$x_{k+1} = x_k^{(2)} - p_k \Gamma_k^{(2)}, \quad \text{(Projection step)} \tag{8c}$$

where $r_k^{(1)} := r_k - q_k\Gamma_k^{(1)}$ and $\beta_k > 0$. Define $r_k^{(2)} = r_k^{(1)} - \beta_k A r_k^{(1)}$ and $r_k^{(3)} = r_k^{(2)} - q_k\Gamma_k^{(2)}$. It can be verified that $r_k^{(1)}, r_k^{(2)}$, and $r_k^{(3)}$ are equal to the residuals at $x_k^{(1)}, x_k^{(2)}$, and $x_{k+1}$, respectively. The coefficients $\Gamma_k^{(1)}$ and $\Gamma_k^{(2)}$ are determined by applying the Galerkin's condition of AM-I:

$$r_k^{(1)} = r_k - q_k\Gamma_k^{(1)} \perp p_k, \quad r_k^{(3)} = r_k^{(2)} - q_k\Gamma_k^{(2)} \perp p_k. \tag{9}$$

Assume $p_k^T q_k \neq 0$. The scheme formulated by (8a)-(8c) gives the $x_{k+1}$ by

$$x_{k+1} = x_k + H_k r_k, \quad H_k = -\frac{p_k p_k^T}{p_k^T q_k} + \beta_k \left(I - \frac{p_k q_k^T}{p_k^T q_k}\right)\left(I - \frac{q_k p_k^T}{p_k^T q_k}\right), \tag{10}$$

leading to the basic Min-AM. The derivation of (10) is in Appendix B.1. It is worth mentioning that the approximated inverse Hessian $H_k$ in (10) is symmetric. Due to the space limit, we give the details of the basic Min-AM in Algorithm 2 in Appendix B. Define $P_k = (p_1, p_2, \ldots, p_k), Q_k = (q_1, q_2, \ldots, q_k)$, we summarize the properties of the basic Min-AM in the next theorem.

**Theorem 1.** *Let $f(x) = \frac{1}{2}x^T A x - b^T x$ where $A$ is SPD and $\{x_k\}$ be the sequence generated by the basic Min-AM for solving (1). $x^*$ is the exact solution. Then, the following properties hold:*
*(i) $\|p_k\|_2 > 0, \text{range}(P_k) = \mathcal{K}_k(A, r_0), \text{range}(Q_k) = A\mathcal{K}_k(A, r_0)$;*
*(ii) $Q_k = -AP_k, p_i \perp q_j (1 \leq i \neq j \leq k)$;*
*(iii) $r_k^{(1)} \perp \text{range}(P_k)$ and $x_k^{(1)} = x_0 + z_k$, where $z_k = \arg\min_{z \in \mathcal{K}_k(A, r_0)} \|x_0 + z - x^*\|_A$.*
*Moreover, if $\|r_k^{(1)}\|_2 = 0$, then $x_{k+1} = x^*$.*

The proof is deferred to Appendix B.2. The property (i) ensures $p_k^T q_k \neq 0$ during the iterations, so $\zeta_k$ and the update (10) are well defined. The property (iii) indicates $x_k^{(1)} = x_k^{CG}$, where $x_k^{CG}$ is the $k$-th iterate of the conjugate gradient (CG) method [32]. Also, due to Proposition 1 in Appendix B.2, the intermediate iterate $\bar{x}_k$ of AM-I($\infty$) satisfies $\bar{x}_k = x_k^{CG}$, so the basic Min-AM is essentially equivalent to the full-memory AM-I when solving unconstrained strongly convex quadratic problems. In this sense, Min-AM can incorporate more historical information than AM(1).

**Remark 2.** *Although Min-AM and CG are equivalent for solving SPD linear systems, their numerical performance has a large difference for general nonlinear programming as shown in our experiments.*

**Remark 3.** *The $H_k$ in (10) is reminiscent of the memoryless BFGS method [4], but the key difference lies in the construction of $p_k$ and $q_k$ in (7). In memoryless BFGS, there are no correction terms ($\zeta_k \equiv 0$), and it is generally not equivalent to BFGS [45] and CG. See Appendix A.3 for more details.*

### 3.2 The restarted Min-AM

For general nonlinear optimization, since global convergence may be unavailable for Min-AM as a counter example exists for AM [41], we introduce the restarting strategy to the basic Min-AM and establish the local convergence analysis. Our purpose is to prove the faster convergence of the restarted Min-AM than that of gradient descent (GD), and sharpen the existing results of AM [59, 21].

**Restarting Strategy.** Let $m_k$ be the number of iterations from the last restart. Min-AM restarts, i.e., setting $m_k = 0$ and $p_k = q_k = \mathbf{0}$, when any of the following conditions is violated:

$$m_k \leq m, \tag{11a}$$

$$|q_k^T p_k| \geq \tau |q_{k-m_k+1}^T p_{k-m_k+1}|, \tag{11b}$$

$$\|\nabla f(x_k)\|_2 \leq \eta \|\nabla f(x_{k-m_k})\|_2, \tag{11c}$$

where $m$ is a constant positive integer, and $0 < \tau < 1, \eta > 0$ are constants. In the interval between two successive restarts, the iterations are the basic Min-AM iterations.

The check (11a) is to limit the number of high-order terms of errors accumulated during iterations. The check (11b) is for the numerical stability of Min-AM since $p_k^T q_k$ appears in the denominators in (10). The check (11c) is to control divergence. The algorithm is shown in Algorithm 1. Let $\mathcal{B}_\rho(x) := \{y \in \mathbb{R}^d | \|y - x\|_2 \leq \rho\}$ denote the ball centered at $x$ with radius $\rho$. We impose the following assumptions on $f$ that are standard in the convergence analysis of quasi-Newton methods [45] and the regularized nonlinear acceleration method [56].

**Assumption 1.** *$f : \mathbb{R}^d \to \mathbb{R}$ is twice Lipschitz continuously differentiable in a local region of a local minimizer $x^*$, and the Hessian matrix $A := \nabla^2 f(x^*)$ is SPD.*

**Algorithm 1** Restarted Min-AM for general nonlinear optimization

---

**Input**: $x_0 \in \mathbb{R}^d, \beta_k > 0, m > 0, \tau \in (0,1), \eta > 0$

1: $p_0, q_0 = \mathbf{0} \in \mathbb{R}^d, m_0 = 0$
2: **for** $k = 0, 1, \ldots,$ until convergence **do**
3:    $r_k = -\nabla f(x_k)$
4:    **if** $m_k > m$ **or** $\|r_k\|_2 > \eta \|r_{k-m_k}\|_2$ **then**
5:       $m_k = 0$
6:    **end if**
7:    **if** $m_k > 0$ **then**
8:       $p = x_k - x_{k-1}, q = r_k - r_{k-1}$
9:       $\zeta_k = (p_{k-1}^{\mathrm{T}} q_{k-1})^{\dagger} p_{k-1}^{\mathrm{T}} q$
10:      $p_k = p - p_{k-1}\zeta_k, q_k = q - q_{k-1}\zeta_k$
11:      **if** $|p_k^{\mathrm{T}} q_k| < \tau |p_{k-m_k+1}^{\mathrm{T}} q_{k-m_k+1}|$ **then**
12:         $m_k = 0$
13:      **end if**
14:    **end if**
15:    **if** $m_k = 0$ **then**
16:       $p_k = q_k = \mathbf{0} \in \mathbb{R}^d$
17:    **end if**
18:    $x_{k+1} = x_k - (p_k^{\mathrm{T}} q_k)^{\dagger} p_k p_k^{\mathrm{T}} r_k + \beta_k \left( I - (p_k^{\mathrm{T}} q_k)^{\dagger} p_k q_k^{\mathrm{T}} \right) \left( I - (p_k^{\mathrm{T}} q_k)^{\dagger} q_k p_k^{\mathrm{T}} \right) r_k$
19:    $m_{k+1} = m_k + 1$
20: **end for**
21: **return** $x_k$

---

From Assumption 1, there exist positive constants $\hat{\rho}, \mu, L,$ and $\hat{\kappa}$ such that for all $x \in \mathcal{B}_{\hat{\rho}}(x^*)$,

$$\mu \leq \frac{y^{\mathrm{T}} \nabla^2 f(x) y}{y^{\mathrm{T}} y} \leq L, \text{ for all } y \in \mathbb{R}^d \text{ and } y \neq \mathbf{0}; \tag{12a}$$

$$\|\nabla f(x) - \nabla^2 f(x^*)(x - x^*)\|_2 \leq \frac{1}{2}\hat{\kappa}\|x - x^*\|_2^2. \tag{12b}$$

Here, $\mu$ is the strong convexity constant, and $L, \hat{\kappa}$ are Lipschitz constants of $\nabla f$ and $\nabla^2 f$ respectively.

With the restarting strategy, we apply a multi-step analysis and has the following convergence theory:

**Theorem 2.** *Suppose that the Assumption 1 holds. Let $\{x_k\}$ be the sequence generated by the restarted Min-AM for solving (1) and $\theta_k = \max\{|1 - \beta_k L|, |1 - \beta_k \mu|\} \leq \theta$ for a constant $\theta \in (0,1)$. Then there exists a positive constant $\rho \leq \hat{\rho}$ such that for any $x_0 \in \mathcal{B}_\rho(x^*)$, we have*

$$\|x_{k+1} - x^*\|_A \leq 2\theta_k \left( \frac{\sqrt{L/\mu} - 1}{\sqrt{L/\mu} + 1} \right)^{m_k} \|x_{k-m_k} - x^*\|_A + \hat{\kappa} \cdot \mathcal{O}(\|x_{k-m_k} - x^*\|_2^2). \tag{13}$$

*Moreover, if $m_k = d$, then $\|x_{k+1} - x^*\|_2 = \hat{\kappa}\mathcal{O}(\|x_{k-m_k} - x^*\|_2^2)$.*

The proof is in Appendix C.1, where a more general theorem (Theorem 6) is given. The result from Theorem 2 suggests that the restarted Min-AM method has a local linear-quadratic convergence rate, and the rate has optimal dependence on the condition number. Compared to the existing results that show the local linear convergence rate of AM is no worse than that of gradient descent (GD) method [59, 16], the bound (13) clearly improves the convergence rate of GD.

**Remark 4.** *Classical nonlinear CG methods [17] are also memory-efficient with recursively modified searching directions, but the step size relies on line-search or Hessian-vector products which can be time-consuming for solving large-scale optimization problems.*

**Remark 5.** *The momentum based methods, e.g. Nesterov's accelerated gradient (NAG) method [44], also achieve the optimal convergence rate if the parameters $\mu$ and $L$ are known. In restarted Min-AM, we give an economical method that approximates the best step size $2/(L + \mu)$ in the next subsection.*

### 3.3 Eigenvalue estimates and the choice of mixing parameter

The mixing parameter $\beta_k$ is critical for AM and AM-I, and an improper choice of $\beta_k$ can lead to divergence [22]. Fortunately, the spectrum of the Hessian can be economically estimated as a

by-product of Min-AM. To better explain the idea, we consider $\min f(x) = \frac{1}{2}x^{\mathrm{T}}Ax - b^{\mathrm{T}}x$ where $A$ is SPD. We apply a projection method [54] to estimate the eigenvalues:

$$v \in \mathcal{K}_k(A, r_0), \quad (A - \lambda I)v \perp A\mathcal{K}_k(A, r_0), \tag{14}$$

where $v \in \mathbb{R}^d$ is the approximate eigenvector sought in the Krylov subspace $\mathcal{K}_k(A, r_0)$, and $\lambda \in \mathbb{R}$ is an eigenvalue estimate, which is known as a generalized Ritz value [42]. Our goal is to compute $\lambda$. Since $\mathrm{range}(P_k) = \mathcal{K}_k(A, r_0)$ as shown in Theorem 1, we can select $v = P_k y$, $y \in \mathbb{R}^k$. Then the projection condition (14) leads to

$$P_k^{\mathrm{T}}AAP_k y = \lambda P_k^{\mathrm{T}}AP_k y. \tag{15}$$

From the construction of $P_k$ and $Q_k$, we can derive the three-term recurrence relations (see details in Appendix D.1): there exists a tridiagonal matrix $\bar{T}_k \in \mathbb{R}^{(k+1) \times k}$, such that

$$AP_k = P_{k+1}\bar{T}_k, \quad AQ_k = Q_{k+1}\bar{T}_k. \tag{16}$$

Then, we have $P_k^{\mathrm{T}}AAP_k = P_k^{\mathrm{T}}AP_{k+1}\bar{T}_k$. By Theorem 1, $P_k^{\mathrm{T}}Ap_{k+1} = -Q_k^{\mathrm{T}}p_{k+1} = 0$. Thus,

$$P_k^{\mathrm{T}}AP_{k+1}\bar{T}_k = P_k^{\mathrm{T}}AP_k T_k, \tag{17}$$

where $T_k \in \mathbb{R}^{k \times k}$ is obtained from $\bar{T}_k$ by deleting its last row. By (15) and (17), we obtain $T_k y = \lambda y$, which indicates that the generalized eigenvalue problem (15) is reduced to solving the eigenvalues of a tridiagonal matrix $T_k$. Moreover, the three elements of the $k$-th column of $\bar{T}_k$ are easy to obtain from the coefficients $\Gamma_k^{(1)}, \Gamma_k^{(2)}, \beta_{k-1}$, and $\beta_k$ on the fly (see details in Appendix D.1):

$$t_k^{(k-1)} = \frac{\Gamma_{k-1}^{(2)}}{\beta_{k-1}(1 - \Gamma_k^{(1)})}, \quad t_k^{(k)} = \frac{1}{1 - \Gamma_k^{(1)}}\left(\frac{1}{\beta_{k-1}} - \frac{\Gamma_k^{(2)}}{\beta_k}\right), \quad t_k^{(k+1)} = -\frac{1}{\beta_k(1 - \Gamma_k^{(1)})}, \tag{18}$$

where $\Gamma_0^{(2)} := 0$. For solving general nonlinear optimization with the restarted Min-AM, suppose that $m_k \geq 1$. We can also construct a tridiagonal matrix $T_k \in \mathbb{R}^{m_k \times m_k}$ starting from the $(k - m_k)$-th iteration with the nonzero elements defined using the coefficients during the iterations (see Definition 2 in Appendix D.2). We use the eigenvalues of $T_k$ as the eigenvalue estimates of the Hessian.

**Theorem 3.** *Suppose that Assumption 1 holds, and $\{x_k\}$ is the sequence generated by the restarted Min-AM for solving (1). Assume that $\beta_j \in [\beta, \beta']$ ($\forall j \geq 0$) for some positive constants $\beta$ and $\beta'$, and there are positive constants $\eta_0, \tau_0$ such that $\|\nabla f(x_j)\|_2 \leq \eta_0 \|\nabla f(x_0)\|_2$ ($0 \leq j \leq k + 1$), $|1 - \Gamma_j^{(1)}| \geq \tau_0$ ($1 \leq j \leq k$); $\lambda$ is an eigenvalue of $T_k$ constructed by restarted Min-AM. Let $\Theta^{(m_k)}$ denote the set of Ritz values computed by an $m_k$-step A-norm based Lanczos algorithm (Algorithm 3 in Appendix D.2) for $\nabla f^2(x^*)$ with starting vector $\nabla^2 f(x^*)(x^* - x_{k-m_k})$. Then there is a positive constant $\rho \leq \hat{\rho}$ such that for any $x_0 \in \mathcal{B}_\rho(x^*)$, we have*

$$\min_{\hat{\lambda} \in \Theta^{(m_k)}} |\hat{\lambda} - \lambda| = \hat{\kappa}\mathcal{O}(\|x_{k-m_k} - x^*\|_2). \tag{19}$$

The proof is in Appendix D.2. Since the Ritz value $\hat{\lambda}$ approximates a true eigenvalue of $\nabla^2 f(x^*)$ [54], Theorem 3 suggests that the restarted Min-AM can give reliable eigenvalue estimates if $\|x_0 - x^*\|_2$ is sufficiently small. At the $k$-th iteration, where $m_k \geq 2$, let $\tilde{\mu}$ and $\tilde{L}$ be the eigenvalues of $T_{k-1}$ with the smallest absolute value and the largest absolute value, respectively. We use $|\tilde{\mu}|$ and $|\tilde{L}|$ to estimate $\mu$ and $L$. Then we can take the mixing parameter as $\beta_k = 2/(|\tilde{\mu}| + |\tilde{L}|)$ to estimate the optimal value $2/(\mu + L)$. The total computational cost is $\mathcal{O}(m_{k-1}^3)$ flops as it requires solving the eigenvalues of $T_{k-1}$ [24]. Since $m_k \ll d$ in practice, this strategy can be a useful and economical option.

**Remark 6.** *The spectrum of Hessian is important for determining the step sizes. Once we obtain the estimates of $\mu$ and $L$ by Min-AM, we can apply them to many first-order methods.*

### 3.4 The stochastic Min-AM

For many applications in deep learning, the large data size prohibits the evaluation of the full gradient, and the stochastic optimization stands out as a solution. Specifically, we consider developing a stochastic version of Min-AM to solve the nonconvex optimization problem:

$$\min_{x \in \mathbb{R}^d} f(x) := \frac{1}{T}\sum_{i=1}^{T} f_i(x), \tag{20}$$

where $f_i : \mathbb{R}^d \to \mathbb{R}$ may be nonconvex. Since $T$ can be extremely large, we sample a mini-batch $S_k \subseteq [T] := \{1, 2, \ldots, T\}$ and use $f_{S_k}(x_k) := \frac{1}{n_k} \sum_{i \in S_k} f_i(x_k)$ to estimate the full gradient $\nabla f(x_k)$, where $n_k := |S_k|$ is the batch size. Then, $r_k := -\nabla f_{S_k}(x_k)$ is an estimate of $-\nabla f(x_k)$.

Due to the noise in the gradient estimates and the nonconvexity of $f$, a direct application of the deterministic Min-AM to (20) may be problematic. For example, the searching direction may be not a descent direction since $H_k$ can be indefinite. Hence, we should introduce some fundamental modifications to the basic Min-AM.

**Regularization.** Inspired by [23], we define the function

$$\Phi(p, q, \delta) = p^{\mathrm{T}} q - \delta(\|p\|_2^2 + \|q\|_2^2). \tag{21}$$

Let $\Phi_k := \Phi(p_k, q_k, \delta_k^{(2)})$ for some $\delta_k^{(2)} > 0$. We substitute $\Phi_k$ for the $p_k^{\mathrm{T}} q_k$ appeared in (10) and define $\rho_k = \Phi_k^\dagger$, i.e. $\rho_k = 0$ if $\Phi_k = 0$ and $\rho_k = \Phi_k^{-1}$ if $\Phi_k \neq 0$. The resulting regularized Min-AM update is $x_{k+1} = x_k + H_k^A r_k$, where

$$H_k^A = -\rho_k p_k p_k^{\mathrm{T}} + \beta_k (I - \rho_k p_k q_k^{\mathrm{T}})(I - \rho_k q_k p_k^{\mathrm{T}}). \tag{22}$$

**Damping.** We propose a specialized damping technique for Min-AM, which interpolates the updates of stochastic gradient descent [50] (SGD) and Min-AM, and is described as

$$x_k^G = x_k + \beta_k r_k, \quad x_k^A = x_k + H_k^A r_k, \quad x_{k+1} = (1 - \alpha_k) x_k^G + \alpha_k x_k^A, \tag{23a}$$

where $x_k^G$ is the SGD update, $x_k^A$ is the Min-AM update, and $\alpha_k$ is the damping parameter. It can be seen that SGD and Min-AM serve as the predicted step, and $x_{k+1}$ is a weighted average of them. Moreover, the inverse Hessian approximation

$$H_k = (1 - \alpha_k)\beta_k I + \alpha_k H_k^A \succeq (1 - \alpha_k)\beta_k I \succ 0 \tag{24}$$

provided that $\alpha_k \in [0, 1)$ and $H_k^A \succeq 0$.

By incorporating damping and regularization, we obtain the stochastic Min-AM (sMin-AM), and the algorithm is summarized in Algorithm 4 in Appendix E. To analyze the convergence properties, we first impose the following assumptions on the objective function $f$, which are the same as [61, 62].

**Assumption 2.** *$f : \mathbb{R}^d \to \mathbb{R}$ is continuously differentiable, and $f(x)$ is lower bounded by a real number $f^{low}$ for $\forall x \in \mathbb{R}^d$. $\nabla f$ is globally $L$-Lipschitz continuous.*

**Assumption 3.** *At each iteration $k$, the gradient estimate $\nabla f_{\xi_k}(x_k)$ satisfies $\mathbb{E}_{\xi_k}[\nabla f_{\xi_k}(x_k)] = \nabla f(x_k)$, $\mathbb{E}_{\xi_k}[\|\nabla f_{\xi_k}(x_k) - \nabla f(x_k)\|_2^2] \leq \sigma^2$, where $\sigma > 0$, and $\xi_k \in [T], k = 0, 1, \ldots$, are independent samples, and the random variable $\xi_k$ (for a given $k$) is independent of $\{x_j\}_{j=0}^k$.*

The diminishing condition about $\beta_k$ is

$$\sum_{k=0}^{+\infty} \beta_k = +\infty, \qquad \sum_{k=0}^{+\infty} \beta_k^2 < +\infty. \tag{25}$$

With chosen $\beta_k$, we choose the $\alpha_k$ and $\delta_k^{(2)}$ of sMin-AM to meet the following conditions:

**Assumption 4.** *Given constants $\mu \in (0, 1)$, $C_1 > 0$, and $C_2 > 0$, we have $\alpha_k \in [0, \min\{C_1 \beta_k^{1/2}, 1 - \mu\}]$, $\rho_k \leq 0$, and $-(\rho_k/\beta_k)\|p_k\|_2^2 - 2\rho_k\|p_k\|_2\|q_k\|_2 + \rho_k^2\|p_k\|_2^2\|q_k\|_2^2 \leq C_2$.*

We establish the convergence results of sMin-AM for nonconvex stochastic optimization and the proofs can be found in Appendix E.2 and Appendix E.3.

**Theorem 4.** *Suppose that Assumptions 2-4 hold and $\{x_k\}$ is the sequence generated by sMin-AM with batch size $n_k \equiv n \leq T$. If $\beta_k \in (0, \frac{\mu}{2L(1+C_2)^2}]$ and satisfies (25), then*

$$\liminf_{k \to \infty} \|\nabla f(x_k)\|_2 = 0 \text{ with probability } 1, \tag{26}$$

*and for all $k$, there exists $M_f > 0$ such that $\mathbb{E}[f(x_k)] \leq M_f$.*

*If $\mathbb{E}_{\xi_k}[\|\nabla f_{\xi_k}(x_k)\|_2^2] \leq M_g$, $\forall k$ where $M_g > 0$ is a constant, we have*

$$\lim_{k \to \infty} \|\nabla f(x_k)\|_2 = 0 \text{ with probability } 1. \tag{27}$$

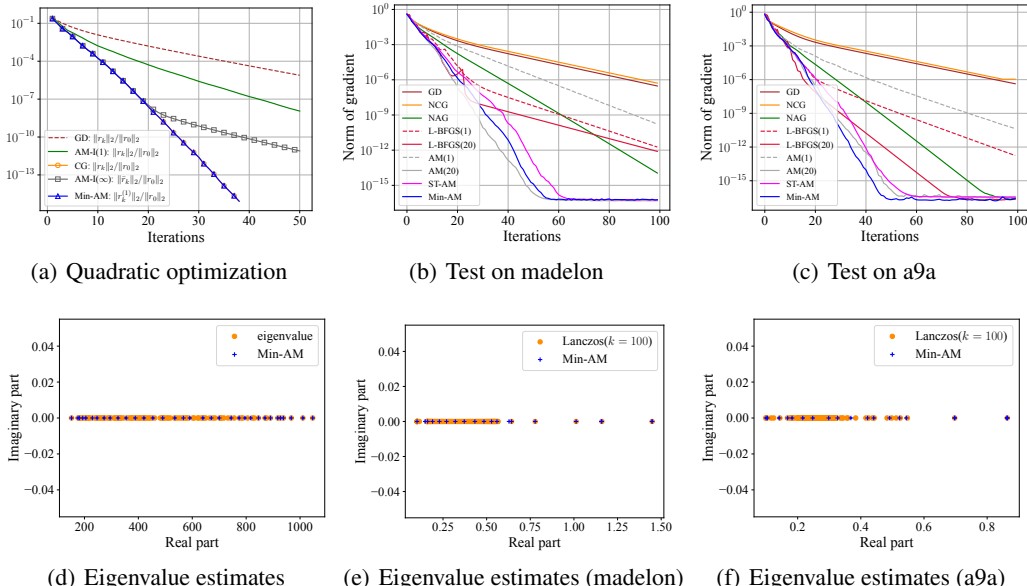

Figure 1: Left (strongly convex quadratic optimization): (a) curves about residuals and intermediate residuals; (d) exact eigenvalues, and eigenvalue estimates from Min-AM. Middle and right (regularized logistic regression on madelon and a9a datasets): (b)(c) $\|\nabla f(x_k)\|_2$ of each method; (e)(f) Ritz values of $\nabla^2 f(x^*)$ by $k$-step Lanczos algorithm, and eigenvalue estimates from Min-AM.

**Theorem 5.** *Suppose that Assumptions 2-4 hold.* $\{x_k\}_{k=0}^{N-1}$ *are the first N iterations generated by sMin-AM with batch size* $n_k \equiv n \leq T$, *and* $\beta_k = \min\{\frac{\mu}{2L(1+C_2)^2}, \frac{\tilde{D}}{\sigma\sqrt{N}}\}$, *where* $\tilde{D}$ *is a problem-independent constant. R is a random variable following* $P_R(k) := \mathrm{Prob}\{R = k\} = 1/N$, $k = 0, \ldots, N-1$. *Then*

$$\mathbb{E}[\|\nabla f(x_R)\|_2^2] \leq \frac{8D_f L(1+C_2)^2}{N\mu^2} + \frac{\sigma}{\mu\sqrt{N}}\left(\frac{4D_f}{\tilde{D}} + \frac{2(\mu^{-1}C_1^2 C_2^2 + L(1+C_2)^2)\tilde{D}}{n}\right), \quad (28)$$

*where* $D_f := f(x_0) - f^{low}$ *and the expectation is taken with respect to R and* $\{S_j\}_{j=0}^{N-1}$. *To ensure* $\mathbb{E}[\|\nabla f(x_R)\|_2^2] \leq \epsilon$, *the number of iterations is* $\mathcal{O}(1/\epsilon^2)$.

**Remark 7.** *The Theorem 4 establishes the global convergence of sMin-AM for solving the problem (20), and Theorem 5 indicates that sMin-AM attains the* $\mathcal{O}(1/\epsilon^2)$ *iteration complexity which is asymptotically optimal for optimization methods based on a stochastic first-order oracle [43].*

## 4 Experiments

We first tested the basic Min-AM in solving strongly convex quadratic optimization (cf. Theorem 1) and the restarted Min-AM on logistic regression (cf. Theorem 2). Then, we applied the sMin-AM to train deep neural networks on CIFAR [38] and ImageNet [19]. Details about the experimental settings and more numerical results can be found in Appendix F.

**Strongly convex quadratic optimization.** As shown in Figure 1(a), the first intermediate residual $r_k^{(1)}$ of Min-AM exactly matches the $k$-th residual of CG, which verifies Theorem 1. Since $X_k^{\mathrm{T}} R_k$ can be ill-conditioned when $k$ is large, it is found that AM-I($\infty$) fails to coincide with CG and Min-AM in the later iterations. The eigenvalue estimates computed by Min-AM also well approximate the true eigenvalues of the Hessian matrix, which is consistent with our analysis in Section 3.3. In Appendix F.1, we also give a discussion about the cost of the eigenvalue estimation procedure.

**Regularized logistic regression.** We conducted the regularized logistic regression on the datasets "madelon" and "a9a" from LIBSVM [14]. The compared methods were GD, nonlinear CG (NCG) with line-search, Nesterov's accelerated gradient (NAG), L-BFGS($m$), AM($m$), and the ST-AM. We

applied the standard $k$-step Lanczos algorithm [24] to compute 100 Ritz values of $\nabla^2 f(x^*)$, in which the smallest and largest ones were used for the setting of $\mu, L$ in NAG. The results in Figure 1 show that Min-AM is competitive with AM(20). It converges much faster than AM(1), L-BFGS(1) (i.e. memoryless BFGS), NCG, and NAG, and outperforms ST-AM. Also, in both datasets, the eigenvalue estimates from Min-AM are accurate enough to approximate the largest and the smallest Ritz values, even though the objective function is non-quadratic. More results including the effect of $\beta_k$ can be found in Appendix F.2.

Table 1: Experiments on CIFAR-10/CIFAR-100. "-" means failing to complete the test in our device.

(a) Final TOP1 test accuracy (mean $\pm$ standard deviation) (%) on CIFAR-10/CIFAR-100.

| Method | CIFAR-10 | | | | | | CIFAR-100 | | |
|---|---|---|---|---|---|---|---|---|---|
| | VGG16 | ResNet18 | ResNet20 | ResNet44 | ResNet56 | WRN16-4 | ResNet18 | ResNeXt | DenseNet |
| SGDM | 93.52±.15 | 94.82±.15 | 92.03±.16 | 93.10±.23 | 93.47±.28 | 94.90±.09 | 77.27±.09 | 78.41±.54 | 78.49±.12 |
| Adam | 92.29±.09 | 93.03±.07 | 91.17±.13 | 92.28±.62 | 92.39±.23 | 92.45±.11 | 72.41±.17 | 73.57±.17 | 70.80±.23 |
| AdaHessian | 93.42±.11 | 94.36±.09 | 91.92±.32 | 92.74±.11 | 92.40±.06 | 94.04±.12 | 76.59±.42 | - | - |
| SAM(1) | 93.35±.12 | 94.95±.10 | 92.10±.05 | 93.15±.23 | 93.44±.19 | 95.02±.32 | 77.06±.48 | 79.12±.40 | 79.58±.29 |
| SAM(10) | 93.59±.11 | 95.17±.10 | **92.43±.19** | 93.57±.14 | 93.77±.12 | **95.23±.07** | 78.13±.14 | 79.31±.27 | 80.09±.52 |
| ST-AM | 93.67±.25 | 95.27±.04 | 92.39±.11 | 93.52±.02 | 93.69±.18 | 95.21±.09 | 77.91±.22 | 79.53±.34 | **80.36±.25** |
| sMin-AM | **94.09±.12** | **95.28±.04** | 92.42±.16 | **93.72±.27** | **93.82±.10** | 95.13±.03 | **78.26±.18** | **79.64±.19** | 79.88±.09 |

(b) The memory and computation cost compared with SGDM. The notations "m","t/e", and "t" are abbreviations of memory, per-epoch time, and total training time, respectively.

| Cost ($\times$ SGDM) | CIFAR-10/ResNet18 | | | CIFAR-10/VGG16 | | | CIFAR-100/ResNeXt50 | | | CIFAR-100/DenseNet121 | | |
|---|---|---|---|---|---|---|---|---|---|---|---|---|
| | m | t/e | t | m | t/e | t | m | t/e | t | m | t/e | t |
| SGDM | 1.00 | 1.00 | 1.00 | 1.00 | 1.00 | 1.00 | 1.00 | 1.00 | 1.00 | 1.00 | 1.00 | 1.00 |
| SAM(10) | 1.73 | 1.78 | 1.00 | 2.51 | 2.59 | 2.59 | 1.30 | 1.16 | 0.58 | 1.16 | 1.19 | 0.60 |
| ST-AM | 1.05 | 1.46 | 0.82 | 1.55 | 1.91 | 1.67 | 1.04 | 1.07 | 0.54 | 1.01 | 1.11 | 0.55 |
| sMin-AM | 1.01 | 1.15 | 0.64 | 1.35 | 1.25 | 0.78 | 1.03 | 1.00 | 0.50 | 1.01 | 1.09 | 0.55 |

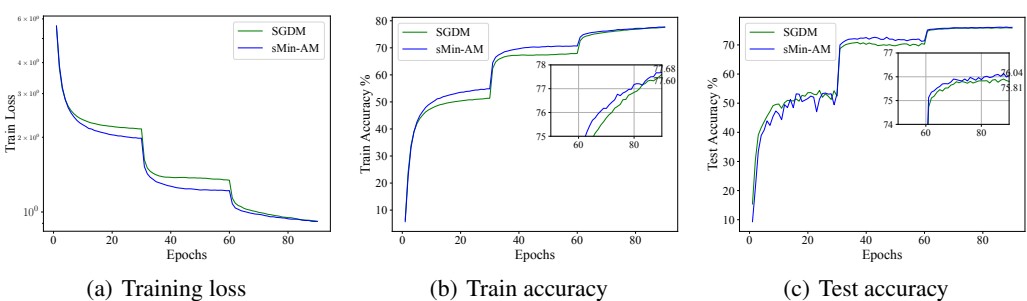

(a) Training loss      (b) Train accuracy      (c) Test accuracy

Figure 2: Experiment of training ResNet50 on ImageNet.

**Experiments on CIFAR.** We applied sMin-AM to train VGG16 [57], ResNet18/20/44/56 [30], WideResNet16-4 [66] (abbr. WRN16-4) on CIFAR-10, and train ResNet18, ResNeXt50 [64], DenseNet121 [34] on CIFAR-100. The compared optimizers were SGDM [49], Adam [36], Ada-Hessian [65], stochastic AM (SAM) [62], and ST-AM [63]. Since the experimental setting was the same as [62, 63], their results were used for reference. Table 1(a) reports the final test accuracy of each optimizer for training 160 epochs. It shows sMin-AM improves SAM(1) and has comparable accuracy to SAM(10). sMin-AM can also use fewer epochs for the training. By setting the accuracy of SGDM as baseline, Table 1(b) shows the memory, per-epoch time, and the total time for an optimizer to achieve a comparable accuracy to SGDM (more details about the number of epochs and final test accuracy can be found in Table 2 in Appendix F.3.2). sMin-AM significantly reduces the memory cost of SAM(10) and the total training time is less than SGDM.

**Experiments on ImageNet.** We trained ResNet50 on ImageNet with SGDM and sMin-AM. Figure 2 shows that the training process of sMin-AM is faster than SGDM. sMin-AM can achieve both higher train accuracy and higher test accuracy in fewer epochs. In our test, we found the memory cost of sMin-AM was 1.04 times that of SGDM, and the per-epoch training time of sMin-AM was 1.01 times that of SGDM. Hence, for neural network training on a larger dataset, sMin-AM can still be competitive with SGDM.

# 5 Conclusion

In this paper, we propose Min-AM, a variant of AM(1) with minimal memory size. Min-AM only stores one recursively modified vector pair and is essentially equivalent to the full-memory Type-I AM in strongly convex quadratic optimization. We establish the convergence properties of Min-AM under deterministic and stochastic settings. For deterministic optimization, we also propose an economical method to estimate the spectrum of Hessian, which can be used to adaptively choose the mixing parameters. The experimental results validate the effectiveness of our methods.

## Acknowledgments and Disclosure of Funding

This work was supported by the National Key R&D Program of China (No. 2021YFA1001300), National Natural Science Foundation of China (No.61925601), National Key Basic Research Program of China (2020YFB0204800), Tsinghua University Initiative Scientific Research Program, National Natural Science Foundation of China (No.11901338), and Huawei Noah's Ark Lab. We thank all anonymous reviewers for their valuable comments and suggestions on this work.

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
