# OpenReview forum: "A Variant of Anderson Mixing with Minimal Memory Size"
_NeurIPS.cc/2022/Conference — NeurIPS 2022 Accept_

### Official Review · Reviewer_wR7v · 2022-07-09

**Rating:** 7
**Confidence:** 5
**Soundness:** 4 excellent
**Presentation:** 2 fair
**Contribution:** 3 good

**Summary:**

The authors present the Min-AM method - an Anderson acceleration technique whose memory is one. The procedure takes advantage of extra projection steps that prevent the loss of some information. The authors show the algorithm's effectiveness by showing its link to conjugate gradient when applied to quadratic functions and its connection with BFGS when looking at the update. Some heuristics are presented to extend the algorithm to nonlinear or stochastic functions, such as restarts, regularization, or adaptive step size.

**Questions:**

I have a few questions concerning the results and conclusion of this paper.
1) After looking at the algorithm and the Conjugate Residual method, it seems there is a connection between those schemes. I wanted to have the authors' opinion about that.
2) In the analysis of the rate of convergence of Theorem 2, you claim that the bound (13) improves on GD. I believe this may not always be the case as a) we do not know how far m_k can go, b) the constant sqrt(L/mu) is larger than one, and c) there is the quadratic factor?

**Limitations:**

See Strengths and weaknesses.

**Strengths And Weaknesses:**

The paper tackles a critical problem of the Anderson Acceleration technique: the memory required to perform an acceleration step. In particular, when solving large-scale optimization problems in a resource-limited machine, the memory requirement may be the bottleneck. The small memory also enables a better analysis of the rate of convergence of the algorithm - as in min-AM, we only have one stored vector rather than a matrix.

From a technical point of view, the paper is solid. It proposes a novel method, makes a connection with classical algorithms, and shows rates of convergence. Moreover, the assumptions made in this paper are rather classical and nonrestrictive.

I have more concerns about the clarity of the paper.
I think some parts could have been improved, especially section 3.1. The section is very dense, and the algorithm's derivation is unintuitive. I suggest the authors spend more time introducing the notations and explaining the purpose of all those modifications (why do 8a, 8b, and 8c help to incorporate more historical information?).
Moreover, the proofs in the appendix are also quite dense. I suggest the author split the big theorem into smaller chunks of lemma to help the reader understand the proofs.

---

> ### Author Response · Authors · 2022-08-02
> **Response to Reviewer wR7v**
>
> Thank you for your support and comments. We hope our replies can address your concerns.
>
> Q1. The clarity of the paper.
> A1. Thank you for your suggestions about the presentation of our paper. We revise the manuscript accordingly.
> (1) The intuition of the additional projection step (8c). It was inspired by the Two-Grid Cycle method [1, Algorithm 13.2] in the multigrid techniques, and (8c) mimics the post-smooth operation in the Two-Grid Cycle procedure.
> (2) Since the concerned objective function in Section 3.1 is quadratic, the definitions of $r_k^{(1)}, r_k^{(2)}, r_k^{(3)}$ are equal to the residuals at $x_k^{(1)}, x_k^{(2)}, x_{k+1}$.
> (3) Compared to AM(1), Min-AM incorporates more information from the previous iterations by recursively updating the stored vector pair. The modified vector pair and the update defined by (8a)-(8c) lead to the high efficiency of Min-AM, i.e., Min-AM is essentially equivalent to the full-memory Type-I AM for solving strongly convex quadratic problems.
>
> Q2. Connection between the method and the conjugate gradient/residual method.
> A2. The Anderson mixing (AM) methods consist of two types of methods: Type-I AM [2] and Type-II AM. It is established in [3] that for solving linear systems, the full-memory Type-I AM and Type-II AM are essentially equivalent to the Arnoldi's method [1, Algorithm 6.4] and GMRES [1, Algorithm 6.9] respectively. For symmetric positive definite (SPD) linear systems, Arnoldi's method and GMRES can be simplified to have short-term recurrences: Arnoldi's method is equivalent to conjugate gradient (CG) method, and GMRES is equivalent to the conjugate residual (CR) method [1, Algorithm 6.20]. Thus, the full-memory Type-I AM and Type-II AM are essentially equivalent to CG and CR respectively, for solving SPD linear systems. However, CG and CR are much more efficient in terms of memory and per-step computational cost, which motivates the development of the proposed Min-AM method that also has short-term recurrences. From Theorem 1, for solving SPD linear systems, Min-AM is essentially equivalent to CG, Arnoldi's method, and the full-memory Type-I AM. Compared to CG,. Min-AM has the advantage that it does not need explicit matrix-vector products to determine step size. The update scheme of Min-AM only depends on the historical iterations, which makes it easier to extend to the nonlinear case.
> Min-AM can also have similar convergence behaviour to CR for solving SPD linear systems,
> due to the relationship between Arnoldi's method and GMRES [1, Section 6.5.7].
>
> Q3. The improvement of Min-AM over GD for general nonlinear optimization in theory.
> A3. We give a refined result of the bound (13) in the revision, and a more general result can be found in Theorem 6 in Appendix C.1 in the revised manuscript. We state it here.
> $$ \\|x_{k+1}-x^*\\|_A \leq \theta_k \min\_\{p \in P\_{m_k}, p(0)=1\}  \\|p(A)(x_\{k-m_k\} - {x^\*})\\|\_A + \hat{\kappa}O(\\|x_\{k-m_k\} -x^\*\\|_2^2),$$
>
> where $P\_{m_k}$ is the space of polynomials of degree not exceeding $m_k$.
> If the sequence $\{x_k\}$ converges to the local minima $x^*$ and
>  $x_0$ is sufficiently close to $x^*$, then the bound is mainly determined by the first-order term, and the convergence rate can be inferred. By choosing $p(A) = (I-\beta_kA)^{m_k}$, we have $\\|p(A)\\|_A = \\|p(A)\\|_2 \leq \theta_k^{m_k} \leq \theta^{m_k} $, which yields that
> $$\theta_k \min\_\{p\in P\_{m_k},p(0)=1\}\\|p(A)(x_\{k-m_k\}-x^\*)\\|_A \leq \theta^{m_k+1} \\|x_\{k-m_k\}-x^\*\\|_A.$$
>
> The convergence rate is no worse than that of gradient descent if $\theta<1$, which has also been established for the limited-memory AM in [4]. When $m_k\geq 1$, we can use Chebyshev polynomials to bound $\min_{p\in P\_{m_k},p(0)=1} \\|p(A)\\|\_2$ and obtain
> $$\\|x_\{k+1\}-x^\*\\|\_A \leq 2\theta_k \left(\frac{\sqrt{L/\mu}-1}{\sqrt{L/\mu}+1}\right)^{m_k} \\|x_\{k-m_k\}-x^\*\\|\_A+\hat{\kappa}O(\\|x_\{k-m_k\}-x^\*\\|\_2^2), $$
>
> which has optimal dependence on the condition number and is significantly better than GD. However, existing results of AM cannot provide such convergence rate guarantee for nonlinear optimization. In practice, we can control the frequency of restarts by $m, \eta, \tau$ in the restarting conditions (11a)-(11c). It is preferable to using moderately large $m_k$ to ensure the fast convergence of Min-AM. Only when the convergence deteriorates due to the high nonlinearity of $f$, more frequent restarts are required.
>
> [1] Yousef Saad. Iterative methods for sparse linear systems. SIAM, 2003.
> [2] Haw-ren Fang and Yousef Saad. Two classes of multisecant methods for nonlinear acceleration. Numerical Linear Algebra with Applications, 16(3):197–221, 2009.
> [3] Homer F Walker and Peng Ni. Anderson acceleration for fixed-point iterations. SIAM Journal on Numerical Analysis, 49(4):1715–1735, 2011.
> [4] Alex Toth and C.T Kelley. Convergence analysis for Anderson acceleration. SIAM Journal on Numerical Analysis, 53(2):805–819, 2015.

---

> > ### Comment · Reviewer_wR7v · 2022-08-08
> > **Reviewer response**
> >
> > I thank the authors for the detailed response to my feedback. I appreciate that the authors have modified the paper accordingly. I encourage the authors to incorporate in their manuscript the connection between the method and the conjugate gradient/residual method.
> >
> > Since the rebuttal is satisfactory, I will keep my current rating.

---

> > > ### Author Response · Authors · 2022-08-09
> > > **Thank you for your support**
> > >
> > > Thanks a lot for your support and comments. We compared the proposed method with the conjugate gradient method in Appendix A.2 in the revised manuscript. We will update the manuscript by further incorporating the connection between the proposed method and the conjugate gradient/residual method.

---

### Official Review · Reviewer_tRei · 2022-07-09

**Rating:** 7
**Confidence:** 4
**Soundness:** 3 good
**Presentation:** 3 good
**Contribution:** 4 excellent

**Summary:**

In this article, the authors introduce a new Anderson mixing method called Min-AM (with restarted and stochastic variants).
The Min-AM method aims at solving the problem of memory required by other Anderson mixing algorithms in the large-scale setting.
It iteratively approximates the inverse of the Hessian by only storing a vector pair $(p_k, q_k)$ capturing the increment in the iterates and in the residuals at previous step. This is why their algorithm achieves minimal memory: $m=1$.
The authors provide
- strong theoretical guarantees concerning the linear-quadratic convergence of the Restarted Min-AM version in Theorem 1
- and convergence guarantees for the stochastic version leading to a $\mathcal{O} (1/ \epsilon^2)$ iteration complexity

Finally, the authors perform 3 sets of experiments on
- quadratic strongly convex synthetic problem : to test the performance of the basic Min-AM algorithm versus other methods (first order or quasi-Newton) and the online eigenvalue estimation (with Lanczos algorithm) required to compute the mixing parameter $\beta_k$
- 2 regularized logistic regression problems on real datasets (a9a and madelon) to compare against other methods as before monitor the spectrum evaluation
- 3 image (large scale) datasets CIFAR10/100 and Imagenet on multiple deep architecture to prove the comparable efficiency of Min-AM with SGD with momentum

~~~~~~~
Score updated after rebuttal (see discussion with the authors), new score: 7

**Questions:**

I'll start with minor comments and typos and then present my questions to the authors:

Minor comments:
- A.1) Notations: I recommend placing real number before vectors in eq (7), that is moving coefficients such as $p_k = \Delta x_{k-1} - \zeta_k p_{k-1}$, same for $q_k$
- A.2) Typo: line 142 "Min-AM and CG are" instead of "is"
- A.3) Algorithm 1: remove "Output: $x \in \mathbb{R}^d$, we don't know what is x and it is redundant with line 21 of the Algorithm
- A.4) line 164 : I think it's good to give names to these 4 quantities $\mu, L, \hat{\rho}$ and $\hat{\kappa}$, What is $\hat{\kappa}$ ?
- A.5) Assumption 2: what is $f^{\text{low}}$?
- A.6) The end of Assumption 3 is strangely written. Again I suggest removing the $\xi_i$
- A.7) Assumption 4, line 248: typo: "are" instead of "is"

Moderate importance comments:
- B.1) In the definition of $r_k^2$ there is an $A$ matrix (line 124) which I find strange as it does not appear in eq (10). I tried to redo the computations and I think there is a mistake here
- B.2) Very strange notations for the Empirical Risk Minimization problem formulation: canonical writing is $f(x) = \frac{1}{n} \sum_{i=1}^n f_i$. I don't think the $\xi_i$ notation is useful, it looks like a random variable but it is not the case.
- B.3) Notations: again, use $T$ iterations instead of $N$
- B.4) line 257: what are the possible values of $k$?

Major comments:
- C.1) Remark 5: I find it unfair to say that NAG requires the knowledge of $\mu$ and $L$ without even mentioning that Min-AM does (through $\theta_k$) and in a more than NAG (as the knowledge of $L$ suffices in practice)
- C.2) line 207: what are $\tilde{\mu}$ and $\tilde{L}$ ?
- C.3) All the plots are in iterations or epochs. Not a single one shows a result in flops or time. Here the compared methods are very different with very different grid-searching schemes and initialization steps. I really wonder what is the performance of Min-AM when the knowledge of the spectrum is not given. I suspect two things: (i) that this initialization computing the spectrum is very costly and (ii) that it is hard to fine-tune the 4 hyperparameters of Stochastic Min-AM
- C.4) There is no explanation, no analysis, no comment of Theorem 4 and 5. The reader directly jumps from one to the other


Questions:
- D.1) Could you please give an intuition about why $H_k$ defined in eq (10) is a (good ?) estimate of the inverse of the Hessian ?
- D.2) Remark 3, line 147: which kind of equivalence are you referring to when you say "it has no equivalence to the fully-memory methods" ? Is this a mistake ?
- D.3) Why do you use pseudo inverse at line 18 of the Algorithm 1 instead of inverse as these quantities are real numbers ?
- D.4) How is the regularization in eq (21) combined with Min-AM. Is it which the basic Min-AM or the restarted one ?
- D.5) How is it possible to have larger time per epoch and smaller total running time for sMin-AM in multiple case of Table 1(b) ?
- D.6) Appendix: Figure 4 : how is it possible to have a condition number smaller than 1 ? By definition, $L > \mu$, so $L/\mu > 1$ and thus cannot equal $10 ^{-2}$ or $10 ^{-8}$.



**Limitations:**

Overall, I am very enthusiastic by the results of the paper, arithmetically and numerically (even though I would like to have more information about practical usage of such methods, especially regarding the initialization and the computation of hyperparameters, sensitivity to them etc)

Yet, I find that the writing and moreover the conference format is degrading the authors' messages. Every result is touched on very quickly (eg the basic and stochastic version of the algorithm are not in the main text, neither the deep comparison with other methods done instead in Appendix A) and thus many explanations are deferred to the appendix (which is ... 33 pages). I would recommend this work to be reorganized and submitted to a journal.

Note that the paper being to dense, I could not go through many proofs.

**Strengths And Weaknesses:**

Strengths:
- this paper provides a valid solution to the memory problem of AM methods which usually require to store large matrices of previous forward differences between past iterates and residuals
- the authors suggest a method to choosing the mixing parameters $\beta_k$ which are based on spectral knowledge of the Hessian
- The restarted and the stochastic variants of the introduced algorithm come with solid convergences proofs.
- The Min-AM method is clearly extended to the stochastic setting making it a valid candidate for training deep architectures
- Experiments, especially on deep net, are very convincing as they show that Stochastic Min-AM competes with SGD + Momentum to train models on CIFAR10, CIFAR100 and Imagenet.

Weaknesses
- Many hyperparameters: the stochastic version for instance method relies on 4 hyperparameters $\alpha_k, \beta_k, \delta_k^{(1)} and \delta_k^{(2)}$ which is not underlined in the paper.
- All the experiments are showing iterations/epochs plots and thus omit many computations required for the evaluation of the spectrum of the Hessian, which is very important as this knowledge is at the core of the setting of the mixing parameters and guarantees good convergence whereas SGD + Momentum would not require this knowledge. Clear time plots with initialization time would have been more informative especially when comparing first order and quasi-Newton methods
- There is no step back taken by authors (even in the conclusion) about weakness or limits or their method (neither theoretically nor numerically)

---

> ### Author Response · Authors · 2022-08-02
> **Response to Reviewer tRei (replies to the weaknesses, minor comments, and moderate importance comments)**
>
> Thank you for your comments and suggestions. We hope our following replies can address your concerns.
>
> Q1. The effects of the hyperparameters of stochastic Min-AM.
> A1.  Since the most critical hyperparameter is the mixing parameter $\beta_k$, we showed the results with different $\beta_k$ in Figure 10 in Appendix B.3.2 in the original manuscript. The results of other hyperparameters are added and shown in Figure 13 in the revised manuscript. It is found that the stochastic Min-AM is not very sensitive to $\alpha_k, \delta_k^{(1)}$, and $\delta_k^{(2)}$ (with $\delta_k^{(2)}\geq 1$). Thus, in practice, we only tune $\beta_k$, and other hyperparameters are set as default.
>
> Q2. The efficiency of the eigenvalue estimation procedure.
> A2. (1) The proposed Min-AM has no initialization cost. The eigenvalue estimation procedure manipulates the coefficients from the previous iterations to construct a tridiagonal matrix $T_k\in \mathbb{R}^{m_k\times m_k}$ (see (18) in Section 3.3) and the main cost is solving the eigenvalues of $T_k$, where $m_k$ is the number of iterations from the last restart. It needs $O(m_k^3)$ flops, which increases with increasing $m_k$. Nevertheless, the eigenvalue estimation can quickly adapt the $\beta_k$ to the optimal choice $2/(\mu+L)$ due to the relationship with Lanczos algorithm, which is efficient for capturing the extreme eigenvalues, and we can terminate the estimation procedure once $\beta_k$ converges.
> In the revised manuscript, we show the efficiency of the eigenvalue estimation in Figure 6 in Appendix F1, where we show the ratio of this additional computational time to the total running time, the convergence behaviour of  $\beta_k$, and the quality of the obtained eigenvalue estimates.
> (2) The total running time on the logistic regression problem was shown in Figure 7 in the original manuscript (Figure 9 in the revised manuscript).
>
> Q3. Limitations of the method.
> A3. Our convergence analysis of Min-AM focuses on the smooth optimization problems. The non-smooth optimization is important, and we leave it as a future work. The proposed method exploits the symmetry of Hessian matrices in optimization problems. Thus, unlike the classical Anderson mixing method, the proposed method is not applicable for solving general fixed-point problems, where the Jacobian matrices can be non-symmetric.
> Due to the space limit, we discussed the above limitations of the proposed method in Appendix C in the original manuscript.  (It can also be found in Appendix G in the revised manuscript.)
>
> Q4. Typos and writing advice.
> A4. Thank you for your suggestions. We correct the typos and add more explanations to the introduced notations.
>  $\hat{\kappa}$ is the Lipschitz constant of $\nabla^2 f$, which affects the error of approximating $f$ via a quadratic function around the local minima $x^*$. $f^{low}$ is a real number that is a lower bound of $f$.
>
> Q5(1). The derivation of Equation (10). How does $A$ disappear in the formula?
> A5(1). The detailed derivation can be found in Appendix A.3 in the original manuscript, or Appendix B.1 in the revised version. The introduced $r_k^{(2)}$ is used to determine the $\Gamma_k^{(2)}$ for the extra projection step (8c). From (9), $\Gamma_k^{(1)} = p_k^T r_k/(p_k^Tq_k)$
> , and
> $\Gamma_k^{(2)} = (p_k^Tr_k^{(2)})/(p_k^Tq_k)$. Here, it follows from $r_k^{(1)}\perp p_k$, the symmetry of $A$, and the relation $q_k = -Ap_k$  that
> $p_k^T r_k^{(2)} = p_k^T(r_k^{(1)}-\beta_kAr_k^{(1)}) = - (Ap_k)^T\beta_k r_k^{(1)}  = \beta_k q_k^Tr_k^{(1)}$. Hence, $\Gamma_k^{(2)} = \beta_kq_k^Tr_k^{(1)}/ (p_k^Tq_k)$, and the formula (10) follows by substituting $\Gamma_k^{(1)}$ and $\Gamma_k^{(2)}$ to (8a)-(8c). The matrix $A$ disappears in (10) due to the relation $q_k = -Ap_k$ in quadratic optimization.
>
> Q5(2). The formulation of the empirical risk minimization problem, and the notations.
> A5(2). Thank you for your suggestions. We have modified the formulation of the empirical risk minimization problem in the revision. In this work, our notations for the stochastic optimization follow those adopted in [1,2]. In the revised manuscript, the notation $\xi_k$ denotes the random variable in $[T] := \\{1,\dots,T\\}$, where $T$ denotes the number of the total data samples. The possible values of $k$ in Theorem 5 are  $0,\dots,N-1$, where $N$ is the number of iterations.
>
> [1] Saeed Ghadimi and Guanghui Lan. Stochastic first- and zeroth-order methods for nonconvex stochastic programming. SIAM Journal on Optimization, 23(4):2341–2368, 2013.
> [2] Xiao Wang, Shiqian Ma, Donald Goldfarb, and Wei Liu. Stochastic quasi-Newton methods for nonconvex stochastic optimization. SIAM Journal on Optimization, 27(2):927–956, 2017.

---

> > ### Comment · Reviewer_tRei · 2022-08-08
> > **Sensitivity to hyperparameters explored and concerned about initialization cost lifted**
> >
> > I thank the authors for explaining the derivation of (10), pointing out that the main hyperparameter to tune is the mixing one and finally, I am now convinced (with Fig 6 (a)) that eigenvalue estimation (performed on the fly as opposed to want I initially understood) does not harm the overall convergence (in term of time).

---

> > > ### Author Response · Authors · 2022-08-09
> > > **Re: Sensitivity to hyperparameters explored and concerned about initialization cost lifted**
> > >
> > > It is our pleasure to address your concerns about our paper. Your detailed comments and valuable suggestions are of great help to the improvement of our work. Thank you!

---

> ### Author Response · Authors · 2022-08-02
> **Response to Reviewer tRei (replies to the major comments)**
>
> Q6(1). The knowledge of $\mu$ and $L$ for Min-AM.
> A6(1). (a) In Theorem 2, we assume $\beta_k$ is chosen such that $\theta_k \leq \theta$, where $\theta<1$ is a constant. This assumption ensures the convergence of Min-AM, which is satisfied by setting $\beta_k = 1$ if the gradient descent to be accelerated has linear convergence. It is also the reason why most theoretical works of Anderson mixing assume the fixed-point map is contractive (e.g., [3,4,5]). In fact, it is still possible for Min-AM to have fast convergence in practice even if $\beta_k$ is improperly set. In Theorem 6 in Appendix C.1 in the revised manuscript, we give a more general result:
> $$\\|x_{k+1}-x^*\\|_A \leq \theta_k \min_{p\in P_{m_k},p(0)=1}\\|p(A)(x_{k-m_k}-x^*)\\|_A+\hat{\kappa}O(\\|x_{k-m_k}-x^*\\|_2^2),$$
> where $P_{m_k}$ is the space of polynomials of degree not exceeding $m_k$. The bound (13) in Theorem 2 can be obtained from this result by using Chebyshev polynomials to bound $\min_{p\in P_{m_k},p(0)=1} \\|p(A)\\|_2$. When $m_k$ is large, the minimization problem on the right-hand side of this bound can dominate the convergence, where fast convergence can be obtained even if $\theta_k>1$. In the revised manuscript, Figure 4(c) for quadratic optimization and Figure 7(a) for logistic regression verify this theoretical finding (we used fixed $\beta_k$ in these tests). In contrast, the gradient descent method diverges if the step size is too large. Hence, the choice of $\beta_k$ for Min-AM is not very stringent.
> (b) For the eigenvalue estimation, the initialization of $\beta_0$ is also not sensitive to $\mu$ and $L$ in practice. We do the tests by choosing large values of $\beta_0$ for Min-AM, which will cause divergence if they are used as step sizes for gradient descent. In the revised manuscript, Figure 7(b) shows that even with badly chosen $\beta_0$, Min-AM still performs well when  $\beta_k$ is adaptively chosen in the subsequent iterations, and the estimates of eigenvalues are still reliable.
>  In summary, the knowledge of $\mu$ and $L$ is not essential for Min-AM in practice.
>
> Q6(2). The $\tilde{\mu}$ and $\tilde{L}$ in Section 3.3.
> A6(2). $\tilde{\mu}$ is the eigenvalue of $T_{k-1}$ of the smallest absolute value, and $\tilde{L}$ is the eigenvalue of $T_{k-1}$ of the largest absolute value. $|\tilde{\mu}|$ and $|\tilde{L}|$ are used to estimate $\mu$ and $L$.
>
> Q6(3). The efficiency of the method measured by running time, the cost of the initialization, and the tuning of hyperparameters.
> A6(3). (1) The results measured by wall time were given in Figure 7 in the original manuscript.
> (2) Min-AM has no initialization cost, and does not require the knowledge of the spectrum in advance. Rather, it can capture useful information of the spectrum based on the historical information. In the revised manuscript, the results in Figure 7(b) and Figure 7(c) suggest that even with very badly chosen $\beta_0$, the eigenvalue estimation procedure still provide accurate enough estimates of $\mu$ and $L$ in the iterations, which accounts for the efficiency of Min-AM since $\beta_k$ is adapted to the optimal value.
> (3) For the stochastic Min-AM (sMin-AM), we only tuned $\beta_0$, and other hyperparameters $\alpha_k, \delta_k^{(1)}, \delta_k^{(2)}$ were set as default. The sensitivity of sMin-AM to these hyperparameters can be found in Figure 12 and Figure 13 in the revised manuscript.
>
> Q6(4). Explanation of Theorem 4 and Theorem 5.
> A6(4). Thank you for your advice. We add Remark 7 in the revision.
>
>
> [3] Alex Toth and C.T. Kelley. Convergence analysis for Anderson acceleration. SIAM Journal on Numerical Analysis, 53(2):805–819, 2015.
> [4] Xiaojun Chen and C.T. Kelley. Convergence of the EDIIS algorithm for nonlinear equations. SIAM Journal on Scientific Computing, 41(1):A365–A379, 2019.
> [5] Claire Evans, et al. A proof that Anderson acceleration improves the convergence rate in linearly converging fixed-point methods (but not in those converging quadratically). SIAM Journal on Numerical Analysis, 58(1):788–810, 2020.

---

> > ### Comment · Reviewer_tRei · 2022-08-08
> > **Comparison of the rate vs GD and more general formulation in Thm 6**
> >
> > I thank the authors for their explanations. I already had a look at your answer to Reviewer wR7v (Q3) about the comparison of your rates vs GD. From what I understood, you claim that the knowledge of $L$ and $\mu$ is not needed as it is only required to set $\theta_k$ (though $\beta_k$) smaller than 1. But even if $\theta_k > 1$, as iterations increase, $m_k$ increases too and the convergence is dominated on the other term independent of $\theta_k$. Am I correct ?
> >
> > Thank you for the other explanations and for taking into account my requests for clarifications in your updated manuscript.

---

> > > ### Author Response · Authors · 2022-08-09
> > > **Re: Comparison of the rate vs GD and more general formulation in Thm 6**
> > >
> > > Thank you for your further comment. The mixing parameter $\beta_k$ along with $L$ and $\mu$ affects the value of $\theta_k$ that appears in the convergence results of the proposed Min-AM method.  With increasing $m_k$ during the iterations, the negative effect of an improperly chosen $\beta_k$ is alleviated since the term $\min_\{p\in P_{m_k}, p(0)=1\}\\|p(A)(x_{k-m_k}-x^*)\\|_A$ is independent of $\theta_k$ and gradually dominates the convergence. Thus, Min-AM can be robust to the setting of $\beta_k$ since $m_k$ is usually not small in practice. In contrast, the gradient descent (GD) method can diverge when its step size is set improperly. Our numerical results also verify this theoretical finding. For example, for solving logistic regression, the result in Figure 7(a) in the revised manuscript shows that Min-AM performs well when using fixed $\beta_k\in\\{2,5\\}$, while GD does not converge if the step size is chosen from $\\{2, 5\\}$.
> > > Hence, the proposed Min-AM method has advantages over GD in terms of the convergence rate and the setting of step size/mixing parameter.

---

> ### Author Response · Authors · 2022-08-02
> **Response to Reviewer tRei (replies to the remaining questions)**
>
> Q7. The reason that $H_k$ defined in Equation (10) is a good estimate of the inverse of the Hessian.
> A7. The explanations can be found in Appendix A.3 in the revised manuscript (or Appendix A2.2 in the original manuscript).
> Define $P_k = (p_1,\dots,p_k), Q_k = (q_1,\dots,q_k)$, where $p_k, q_k$ are the vectors in Min-AM, and define
> $$H_k^{A} = -P_k(P_k^TQ_k)^{-1}P_k^T+\beta_k(I-P_k(P_k^TQ_k)^{-1}Q_k^T)(I-Q_k(P_k^TQ_k)^{-1}P_k^T).$$   We call the iterations defined by $x_{k+1} = x_k+H_k^{A}r_k$ as Scheme A, where $r_k = -\nabla f(x_k)$. Consider strongly convex quadratic problems.
> With the properties in Theorem 1, it can be verified that
> $H_kr_k = H_k^A r_k$. Note that when fixed $\beta_k$ is used, i.e., $\beta_k\equiv \beta$, where $\beta$ is a constant,   $H_k^A$ can be recursively constructed:
> $$H_k^A = -\frac{p_kp_k^T}{p_k^Tq_k}+(I-\frac{p_kq_k^T}{p_k^Tq_k})H_{k-1}^A(I-\frac{q_kp_k^T}{p_k^Tq_k}), $$
> with $H_0^A : = \beta I$.
> This formula suggests that $H_k^A$ solves
> $\min\_{H}\\|H-H_{k-1}^A\\|\_{F(W)}$,   s.t.   $Hq\_k = -p\_k,   H=H^T$,
> where $\\|\cdot\\|\_{F(W)}$ is the weighted Frobenius norm (i.e., $\\|X\\|\_{F(W)}:=\\|W^{1/2}XW^{1/2}\\|\_F$ for a matrix $X\in\mathbb{R}^{d\times d}$) with $W$ satisfying $Wp_k = -q_k$.
> This problem is the same as the problem in BFGS for updating the approximate inverse Hessian, except that BFGS uses $H(\nabla f(x_k)-\nabla f(x_{k-1})) = x_k – x_{k-1}$. Since $H_k^A Q_k=-P_k$,  we know $H_k^A$ will finally be equal to the inverse Hessian when $k=d$.  Also, from $H_kr_k = H_k^{A}r_k$, it follows that the iterations of Min-AM are identical to those of Scheme A. In this sense, we say Min-AM well approximates the inverse Hessian using historical information.
> Also, Min-AM is much more memory-efficient than Scheme A since it does not need to store the whole historical vector pairs ($P_k, Q_k \in\mathbb{R}^{d\times k}$).
>
> Q8. The Remark 3 about the memory-less BFGS.
> A8. Memory-less BFGS only stores the most recent vector pair $x_k-x_{k-1}, \nabla f(x_k)-\nabla f(x_{k-1})$, and discards the former historical information. Hence, memory-less BFGS is not equivalent to the full-memory quasi-Newton methods (BFGS, AM, and AM-I) in general.
>
> Q9. Why is the pseudo inverse used in Line 18 of Algorithm 1?
> A9. We use pseudo inverse because $p_k^Tq_k$ can be zero, and the inverse is invalid in this case.
>
> Q10. How is the regularization in Equation (21) combined with Min-AM. Is it the basic Min-AM or the restarted one ?
> A10. It is the basic Min-AM.
> Details can be found in Algorithm 4.
>
> Q11. How is it possible to have larger per-epoch time but smaller total running time for stochastic Min-AM in Table 1(b)?
> A11. Since sMin-AM needs fewer epochs to achieve a comparable accuracy to that of SGDM, the total running time is reduced though sMin-AM is more costly in each iteration. The complete results, including the total training epochs and final accuracy, are given in Table 2 in Appendix F.3.2 in the revised manuscript (or Table 2 in Appendix B.3.2 in the original manuscript).
>
> Q12. The condition number $L/\mu$ in Figure 5.
> A12. Thank you for pointing it out. The correct forms are $L/\mu =10^2$ and $L/\mu = 10^8$. We have corrected it in the revision.
>
> Q13. The organization of the paper.
> A13. We reorganize the contents of the paper including the appendix in the revision.  The main changes in the revised manuscript can be found in our general response to all the reviewers.

---

> > ### Comment · Reviewer_tRei · 2022-08-08
> > **Better insight given for $H_k$ and Remark related to L-BFGS**
> >
> > Thank you for the insight given concerning the estimate of the Hessian in your framework.
> >
> > A11) Of course, I don't know why I thought you were performing a fixed and common number of iterations for all methods...
> >
> > The updated version of Remark 3 is clearer, thank you.

---

> > > ### Author Response · Authors · 2022-08-09
> > > **Re: Better insight given for $H_k$ and Remark related to L-BFGS**
> > >
> > > Thank you again for your suggestions about the presentation of our paper. We will continue working on the clarification of our work.

---

### Official Review · Reviewer_3SM6 · 2022-07-11

**Rating:** 5
**Confidence:** 3
**Soundness:** 3 good
**Presentation:** 3 good
**Contribution:** 2 fair

**Summary:**

This paper proposes a new variant of the Anderson mixing algorithm with minimal memory size.

**Questions:**

1. Page 9, upper panel of Table 1: Compared to ST-AM, the numerical results presented in Table 1 are not  statistically significant (except for the case of VGG16).  It is hard to justify the significance of the method in test accuracy improvement.

2. Page 9, lower panel of Table 1: It seems that the improvement is marginal for large-scale problems such as CIFAR100. Please comment on how the performance of the algorithm changes with the scale of the problem.

**Limitations:**

Yes.

**Strengths And Weaknesses:**

The theoretical development for the proposed algorithm seems solid, but the numerical improvement seems incremental.

---

> ### Author Response · Authors · 2022-08-02
> **Response to Reviewer 3SM6**
>
> Thank you for your comments. We hope our following replies can address your concerns and better clarify our contributions.
>
> Q1. The significance of improvement in test accuracy for training neural networks.
> A1. Compared to ST-AM, the proposed stochastic Min-AM (sMin-AM) method uses smaller memory footprint and computational cost, while achieving higher test accuracy in 7 networks of the total 9 networks in Table 1(a). To further compare the test performance between sMin-AM and ST-AM, we conduct the tests of training ResNet50 on ImageNet and adversarial training on CIFAR10/WideResNet34-10 and CIFAR100/DenseNet121, and the results are given in Section F.3.3 and Section F.3.4 in the revised manuscript. In the ImageNet/ResNet50 test, sMin-AM achieves 0.45% higher test accuracy than ST-AM. In the adversarial training, sMin-AM also  improves the test accuracy of ST-AM on the corrupted datasets, where the improvement on CIFAR10/WideResNet34-10 is more significant (1.06% higher FGSM-attack test accuracy, 1.21% higher PGD20-attack test accuracy, and 1.17% higher C\&W$_{\infty}$ test accuracy).
> In addition, we would like to mention that AM has been successfully applied to many tasks, such as accelerating EM algorithm [1], reinforcement learning [2], and solving minimax problems [3]. However, the memory size in AM is usually heuristically set in practice. Although the ST-AM has reduced the memory size to 2, it is still of theoretical interest if we can further reduce the memory size to the minimal size with theoretical convergence analysis. Moreover, we give an adaptive strategy to choose the mixing parameter by deeply exploring the properties of the proposed method. Consequently, we hope our work makes a more important attempt to analyze extrapolation methods based on AM.
>
> Q2. How does the memory and computation cost of the algorithm change with the scale of the problem.
> A2. (1)  Given a specific network, the cost is composed of two parts: (a) updating the network parameters by the optimizer; (b) other necessary computations, such as the forward and backward propagations of the neural network, data transfers between memory and disks, etc., where additional memory and processing time are required. If the cost of Part (b) only occupies a small proportion of the total cost, the memory footprint and computational cost of the optimizer will be of great importance.
> (Effect of batch size) In Figure 16 in the revised manuscript, we show the memory and per-epoch running time of SAM(10), ST-AM, and sMin-AM for training ResNeXt50 on CIFAR-100 with different batch sizes. It is found that when smaller batch size is used, the improvement of sMin-AM over SAM(10)/ST-AM becomes more significant in terms of computational cost and memory cost.
> (Effect of model size) In Table 3 in the revised manuscript, we show the memory and per-epoch running time of SAM(10), ST-AM, and sMin-AM for training ResNet18/ResNet50 on CIFAR-10, where the batch size is 16 so that the cost of Part (b) is not large. The results indicate that for a network of larger scale, sMin-AM is more advantageous than SAM(10)/ST-AM in terms of memory cost.
> Hence, these numerical results are consistent with our analysis for the cost of an optimizer.
> (2) The difference between CIFAR10 and CIFAR100 has minor effect on the cost. For a given network architecture, e.g. VGG16, its implementations for CIFAR10 and CIFAR100 mainly differ in the dimension of the last linear layer, whose parameters only occupy a very small portion of the whole parameters for a large neural network. Therefore, the costs (memory and per-epoch running time) on CIFAR-10 and CIFAR-100 are roughly the same, as shown in Table 4  in the revised manuscript.
>
> [1] Nicholas C Henderson and Ravi Varadhan. Damped Anderson acceleration with restarts and
> monotonicity control for accelerating EM and EM-like algorithms. Journal of Computational
> and Graphical Statistics, 28(4):834–846, 2019.
> [2] Ke Sun, et al. Damped Anderson mixing for deep reinforcement learning: Acceleration, convergence, and stabilization. Advances in Neural Information Processing Systems, 34, 2021.
> [3] Huan He, et al. GDA-AM: On the effectiveness of solving min-imax optimization via Anderson mixing. International Conference on Learning Representations. 2021.

---

### Official Review · Reviewer_R1xQ · 2022-07-20

**Rating:** 7
**Confidence:** 3
**Soundness:** 3 good
**Presentation:** 2 fair
**Contribution:** 3 good

**Summary:**

The contribution proposes Min-AM, which is the minimum memory variant of Anderson Mixing/acceleration well-known in fixed-point recursions.

**Questions:**

+ Is it possible to clarify some figures as commented above? In particular, the authors may want to plot the error between the corresponding pairs of eigenvalues to show the results more clearly.

+ What is/are the limitation(s) of the proposed method?

**Limitations:**

I did not find any discussion on this.

**Strengths And Weaknesses:**

+ The main ideas and results are well-organized and presented with clarity.

+ In this reviewer's understanding, the main/novel contribution of this work are contained in Sections 3.2 and 3.3.

+ The Appendix is very detailed and provides the technical support behind the statements in the main text.

+ Fig. 1(d),(e),(f) are illegible due to the lack of correspondence info between the pluses and circles.

---

> ### Author Response · Authors · 2022-08-02
> **Response to Reviewer R1xQ**
>
> Thank you for your support and comments. We hope our replies can address your concerns.
>
> Q1. Clarification of Figure 1(d), 1(e), 1(f).
> A1. We add more explanations in the revision to clarify the meanings of these figures.
> As described in Section 3.3, Min-AM constructs a tridiagonal matrix $T_k$ using the coefficients during the iterations, and uses the eigenvalues of $T_k$ to estimate the eigenvalues of the Hessian matrix. In Figure 1(d), the circles are the exact eigenvalues of the Hessian matrix;  the pluses are the eigenvalue estimates computed by Min-AM, i.e. eigenvalues of $T_k$.
> In the tests of logistic regression, since the Hessian matrix is difficult to obtain, we used the Lanczos algorithm to compute 100 Ritz values of $\nabla^2 f(x^*)$, where $x^*$ is the minima.  From the classical analysis of Lanczos algorithm [1, Theorem 6.4], the Ritz values can well approximate the true eigenvalues of $\nabla^2 f(x^*)$. In Figure 1(e) and Figure 1(f), the circles are the Ritz values, and the pluses are the eigenvalue estimates computed by Min-AM.
> These results verify our Theorem 3 in Section 3.3, and indicate that Min-AM can provide reliable eigenvalue estimates, which accounts for effectiveness of the proposed strategy to adaptively choose $\beta_k$ based on the eigenvalue estimates.
>
> Q2. Limitations of the method.
> A2. Our convergence analysis of Min-AM focuses on the smooth optimization problems. The non-smooth optimization is important, and we leave it as a future work. The proposed method exploits the symmetry of Hessian matrices in optimization problems. Thus, unlike the classical Anderson mixing method, the proposed method is not applicable for solving general fixed-point problems, where the Jacobian matrices can be non-symmetric.
> Due to the space limit, we discussed the above limitations of the proposed methods in Appendix C in the submitted manuscript.  (It can also be found in Appendix G in the revised manuscript.)
>
> [1] Yousef Saad. Numerical methods for large eigenvalue problems (revised edition). SIAM, 2011.

---

### Author Response · Authors · 2022-08-02
**Response to all the reviewers**

Dear reviewers,

We are grateful for your valuable comments and constructive suggestions. We tried our best to revise the paper and hope the refined theoretical results, additional experimental results, additional  discussions, and the improved presentation can better clarify our contributions and demonstrate the advantages of the proposed Min-AM methods. We summarize the main changes as follows.
1. Refined theoretical results.
We refine the convergence bound (13) in Theorem 2, and give a more general convergence result in Theorem 6 in Appendix C.1. The assumption of Theorem 3 is also relaxed based on Theorem 6.

2. Additional experimental results.
We provide more comprehensive tests about the robustness of the proposed method/strategy to initialization/hyperparameters, and cost of the computations in Appendix F. The additional experiments in Appendix F.3.3 and Appendix F.3.4 compare the test performance of Min-AM with ST-AM for training ResNet50 on ImageNet, and for adversarial training on CIFAR10/WideResNet34-10 and CIFAR100/DenseNet121.

3. Additional discussions.
For theory, we add Remark 7 about the convergence results in stochastic optimization, and Remark 8 and Remark 9 about the convergence theory in general nonlinear optimization.
For the experiments, we add discussions about the cost of the eigenvalue estimation procedure (Appendix F.1), the robustness of the proposed method/strategy to initialization (Appendix F.2) and hyperparameters (Appendix F.3.2), and the computation and memory cost (Appendix F.3.2).

4. Improved presentation.
(1) We add more explanations of the methods, including the intuition of introducing an extra projection step to the original Anderson mixing scheme (Section 3.1),  the description of  introduced notations, e.g., $r_k^{(1)}, r_k^{(2)}, r_k^{(3)}$ in Section 3.1, and the construction of $T_k$ in Definition 2 in Appendix D.2.
(2) We reorganize the contents in the appendix, which now contain
 comparisons with conventional methods (Newton's method, conjugate gradient method, BFGS, momentum-based method, and related variant of Anderson mixing methods),
details of the methods (basic Min-AM for strongly convex quadratic optimization, restarted Min-AM for general nonlinear optimization, eigenvalue estimation, and stochastic Min-AM for stochastic optimization), experimental details, and limitations.
(3)The proofs of the theorems are also reorganized to make them more comprehensible: The convergence result of the basic Min-AM from Theorem 1 is summarized in Corollary 1, which is the base for the analysis of restarted Min-AM; the proof of Theorem 2 is decomposed into several steps (Definition 1, Lemma 1,  and Theorem 6), and Lemma 1 is reused in the proof of Theorem 3; the proofs of Theorem 4 and Theorem 5 begin with several useful lemmas.
(4) For the experimental details, the purpose of each numerical test is further clarified in the corresponding paragraph in Appendix F.

We hope the revision and the following replies can address your concerns and help you better evaluate our work.

Thank you!

Yours faithfully,

The Authors.

---

> ### Comment · Reviewer_tRei · 2022-08-08
> **Revised score**
>
> Dear authors,
>
> thank you for you very detailed answers. Even though I still find the content of your paper very dense and maybe had to grasp for non specialists, I could see improvements in rewriting and reformatting your manuscript (especially the appendix which is clearer now).
>
> I think this paper could be very interesting for the optimization community working on limited-memory Newton-like methods. Thus I decided to raise my score to 7 as most of my concerns (technical computation in (10), intuition behind the Hessian estimate, cost of computing the eigenvalues etc) were solved.
>
> Best regards,
>
> Reviewer tRei

---

> > ### Author Response · Authors · 2022-08-09
> > **Thank you for your support**
> >
> > Dear Reviewer tRei,
> >
> > Thank you very much for your support and raising the score. Our paper benefits a lot from your detailed comments and valuable suggestions. We really appreciate it.
> >
> > Yours faithfully,
> >
> > The Authors.

---

### Meta-Review · Area_Chair_MCoj · 2022-08-25

**Recommendation:** Accept
**Confidence:** Certain

**Metareview:**

All reviewers were positive about this paper and the overall impression is very good - accept.

**Award:**

No

---

### Decision · Program_Chairs · 2022-09-14

Accept